# Congenital Zika Virus Infection Impairs Corpus Callosum Development

**DOI:** 10.3390/v15122336

**Published:** 2023-11-28

**Authors:** Raissa Rilo Christoff, Jefferson H. Quintanilha, Raiane Oliveira Ferreira, Jessica C. C. G. Ferreira, Daniel Menezes Guimarães, Bruna Valério-Gomes, Luiza M. Higa, Átila D. Rossi, Maria Bellio, Amilcar Tanuri, Roberto Lent, Patricia Pestana Garcez

**Affiliations:** 1Institute of Biomedical Sciences, Federal University of Rio de Janeiro, Rio de Janeiro 21941-590, RJ, Brazil; raissa.rilo@gmail.com (R.R.C.); jessicadecassiacfg@gmail.com (J.C.C.G.F.); danielmgui@gmail.com (D.M.G.);; 2Human Genome and Stem Cell Research Center, Department of Genetics and Evolutionary Biology, Institute of Biosciences, University of São Paulo, São Paulo 05508-090, SP, Brazil; 3Institute of Medical Biochemistry Leopoldo de Meis, Federal University of Rio de Janeiro, Rio de Janeiro 21941-902, RJ, Brazil; 4Department of Genetics, Institute of Biology, Federal University of Rio de Janeiro, Rio de Janeiro 21941-902, RJ, Brazil; 5Institute of Microbiology Paulo de Góes, Federal University of Rio de Janeiro, Rio de Janeiro 21941-902, RJ, Brazil; mariabellioufrj@gmail.com; 6Institute D’Or for Research and Education, Rio de Janeiro 2281-100, RJ, Brazil

**Keywords:** axon growth, axon guidance, brain malformation, congenital Zika syndrome, corticogenesis, callosal dysgenesis, microcephaly, ZIKV

## Abstract

Congenital Zika syndrome (CZS) is a set of birth defects caused by Zika virus (ZIKV) infection during pregnancy. Microcephaly is its main feature, but other brain abnormalities are found in CZS patients, such as ventriculomegaly, brain calcifications, and dysgenesis of the corpus callosum. Many studies have focused on microcephaly, but it remains unknown how ZIKV infection leads to callosal malformation. To tackle this issue, we infected mouse embryos in utero with a Brazilian ZIKV isolate and found that they were born with a reduction in callosal area and density of callosal neurons. ZIKV infection also causes a density reduction in PH3+ cells, intermediate progenitor cells, and SATB2+ neurons. Moreover, axonal tracing revealed that callosal axons are reduced and misrouted. Also, ZIKV-infected cultures show a reduction in callosal axon length. GFAP labeling showed that an in utero infection compromises glial cells responsible for midline axon guidance. In sum, we showed that ZIKV infection impairs critical steps of corpus callosum formation by disrupting not only neurogenesis, but also axon guidance and growth across the midline.

## 1. Introduction

Zika virus (ZIKV) has gained worldwide attention since its devastating impact on brain development in humans was identified [1]. The virus infects neural progenitor cells in vitro and in vivo, impairing the cell cycle, causing cell death, and leading to microcephaly [2,3,4]. Although microcephaly is the main feature of ZIKV infection, other brain abnormalities have been identified [5,6]. The set of signs present in affected newborns whose mothers were infected during pregnancy is now known as congenital Zika syndrome (CZS). Alongside microcephaly, CZS patients also display ventriculomegaly, brain calcifications, and corpus callosum dysgenesis [7,8].

The corpus callosum is the main telencephalic commissure of placental mammals, connecting the two cerebral hemispheres and contributing to the proper functioning of different sensorimotor and cognitive abilities [9]. Corpus callosum dysgenesis is a brain malformation characterized by the absence or size reduction in the commissure, with an abnormal formation of ectopic bundles [10,11,12]. In typical neurodevelopment, callosal projection neurons arise from neural progenitor cells and migrate to their final position on the cortical plate where axons are extended to a contralateral target following guidance molecules, including those secreted by midline glial cells [13]. In mice, midline glial populations are generated from radial glial progenitors between embryonic days E13 and E17, and express the astroglial marker GFAP. These structures are known as the glial wedge, indusium griseum glia, and midline zipper glia, positioned close to or at the boundary between the hemispheres [13]. Defects of midline structures compromise axon guidance of callosal neurons, resulting in corpus callosum malformations [14,15].

During corticogenesis, ZIKV early infection disrupts radial glial progenitor cells by altering their proliferation, as shown by the reduction in proliferation markers such as Ki67 and PH3. Also, viral infection reduces intermediate progenitors that are responsible for cerebral cortex expansion [2,16]. This scenario leads to a reduced cortical thickness from the loss of neuronal cells of different cortical layers. In addition to disrupting the proliferation of progenitor cells, ZIKV infection causes cell death via apoptosis and more severe responses, such as pyroptosis [17]. In later infection, ZIKV targets progenitor cells that will give rise to the oligodendrocytes, causing impairment in proliferation and differentiation [18]. Primarily, ZIKV impairs proliferation, but it can also infect different types of neural cells such as astrocytes, postmitotic neurons, and microglia [19,20,21,22]. Additionally, to its effects on neurogenesis, infection of astrocytes and microglia causes the secretion of pro-inflammatory cytokines that ultimately contribute to neuroinflammation [19]. Since ZIKV is capable of infecting different types of glial cells and progenitors, both at the early and late stages of brain development, it is conceivable that ZIKV infection indirectly disturbs corpus callosum development.

Although many studies have shown the impact of ZIKV in neurogenesis and its link to microcephaly, it is unknown how ZIKV interferes with corpus callosum development. Therefore, this study aimed to identify the cellular mechanisms that could lead to corpus callosum dysgenesis in ZIKV infection in an in utero animal model. Here, we provide evidence that ZIKV infection during corticogenesis causes corpus callosum developmental abnormalities by not only reducing the callosal neuron production but also impairing axon guidance and outgrowth.

## 2. Materials and Methods

### 2.1. Experimental Design

We studied the effect of in utero ZIKV infection on the different steps of corpus callosum development. To specifically target the birth of callosal neurons, we injected at embryonic day (E) 15, the peak of neurogenesis. To target both callosal neurons and midline glia birth, we performed an early injection at E13. Both male and female mice were analyzed since analyses were made at the embryonic and postnatal stages. For histological and immunohistochemical analysis, a sequence of 3 brain slices were used per animal. For in vitro analysis, 3 independent experiments were performed. More details are described below.

### 2.2. Animals and ZIKV Infection

Pregnant Swiss mice were housed at the Animal Care Facility of the Microbiology Institute of the Federal University of Rio de Janeiro, and maintained in standard animal housing with food and water ad libitum and circadian cycles of 12 h light/12 dark. Pregnant dams were anesthetized, and the uterus was exposed to handle the embryos carefully. A volume of 1.5 μL of a Brazilian ZIKV isolate (Recife/Brazil, ZIKV PE/243, accession no: KX197192.1) or MOCK was injected into one side of the lateral ventricle of the embryonic mouse brain. On E15, the mice were injected with 10^4^ PFU (plaque-forming units) of ZIKV brain tissues and were harvested either E18 or postnatal day 4 (P4). On E13, the mice were injected with 30 PFU of ZIKV and harvested at either E16, E18, or P0.

### 2.3. Virus Detection

Viral RNA was extracted from primary neuronal culture supernatant after four days in vitro (4DIV) with 3 days post-infection (dpi), using the RNeasy Plus Mini Kit (QIAGEN, Venlo, The Netherlands), following the recommendations of the manufacturer. To determine the viral load of these samples, reverse transcription of viral RNA followed by quantitative PCR was performed with GoTaq Probe 1-Step RT-qPCR System (Promega, Madison, WI, USA) on a 7500 Real-Time PCR System (Applied Biosystems, Foster City, CA, USA), using the primers and probe described by Lanciotti et al. (2008) [23]. ZIKV RNA copies were calculated via interpolation onto a standard curve composed of eight 10-fold serial dilutions of a synthetic ZIKV RNA based on the region targeted by the set of primers and probe. The ZIKV quantification was expressed as ZIKV RNA copies per gram of tissue. The quantification of infectious particles was performed via a plaque assay. Brain tissue was mechanically homogenized with ultra turrax T10 (ika) in 1 mL of DMEM supplemented with 2% fetal bovine serum and 2× penicillin/streptomycin. Brain tissue homogenates were centrifuged at 3400× *g* for 5 min. The clarified supernatants were serially diluted and incubated with confluent monolayers of Vero cells. Virus titers were determined by a plaque assay performed on Vero cells. Virus stocks or samples were serially diluted and adsorbed to confluent monolayers. After 1 h, the inoculum was removed, and cells were overlaid with a semisolid medium composed of alpha-MEM (GIBCO, Miami, FL, USA) containing 1.25% carboxymethyl cellulose (Sigma Aldrich, Burlington, MA, USA) and 1% FBS (GIBCO, Miami, FL, USA). The cells were further incubated for 5 days once they were fixed in 4% formaldehyde. Cells were stained with 1% crystal violet in 20% ethanol for plaque visualization. Titers were expressed as PFU per gram.

### 2.4. Primary Neuronal Culture and Infection

Pregnant dams were euthanized via cervical dislocation after anesthesia. E14 brains were harvested and dissected, followed by mechanical dissociation of the cerebral cortex with a micropipette into a cell suspension. The dissociation of cortical tissue was performed in Neurobasal at a final volume of 1 mL. Then, 100,000 cells were added per plate with the Neurobasal culture medium supplemented with 2% B27 and antibiotics, then kept in the incubator under 5% CO_2_ at 37 °C. After 1DIV, cortical neurons were exposed to ZIKV (MOI 1) or MOCK. After 1 h, the original culture medium was returned to the culture, and the neurons were incubated for a further 3 days (3 dpi). With 4DIV, the culture was fixed with 4% paraformaldehyde for further immunohistochemical analysis.

### 2.5. Immunohistochemistry and Imaging

Embryonic brains were fixed with 4% paraformaldehyde overnight, cut coronally at 70 µm with a vibratome (VT1000S, Leica, Berlin, Germany), and stored in 0.01% PBS. After standard antigenic retrieval, coronal sections were permeabilized with 0.1% Triton X-100 (Sigma-Aldrich, Burlington, MA, USA) and incubated with 3% bovine serum albumin (Sigma-Aldrich, Burlington, MA, USA) for 2 h. Next, the following primary antibodies were incubated overnight: mouse anti-SATB21:200 (GenWay 20-372-60065, San Diego, CA, USA); rabbit anti-PH3 1:500 (Millipore 06-570); mouse anti-GFAP 1:200 (Sigma G3893); rabbit anti-TBR2 1:500 (Abcam AB23345, Cambridge, UK); rabbit anti-Caspase3 1:300 (Cell signaling 9664, Danvers, MA, USA); anti-IBA1 1:1000 (Wako 019-19741, Monza, Italy); and mouse anti-NS1 1:10. Subsequently, the samples were washed with PBS and incubated with secondary antibodies, including goat anti-rabbit Alexa Fluor 488 1:500 AP132JA4 (Millipore, Toronto, ON, Canada) and goat anti-mouse Alexa 546 1:500 AP192SA6 (ThermoFisher Scientific, Waltham, MA, USA). The nuclei were stained with DAPI (0.5 mg/mL) for 20 min. Images were acquired with a TCS SP8 confocal microscope (Leica, Berlin, Germany) with an oil immersion 20× objective of high numerical apertures (NA). Analyses of the acquired images were carried out using Fiji software 2.9 version.

### 2.6. Immunocytochemistry and Quantitative Analyses

Three different experiments were carried out in duplicates for further analysis. Immunocytochemistry was carried out similarly after vibratome brain sections. The following primary antibodies were used: rabbit SATB2 1:200 (Abcam AB34735); and mouse Tuj1 1:100 (Millipore MAB1637). For evaluation of the axonal length, approximately 30 cells per well were equally analyzed under experimental conditions. To establish a comparison criterion, the largest axons of positive DAPI/SATB2/TUJ1 neurons were chosen to be analyzed in the two experimental groups. The axonal length was measured on ImageJ, outlining the extension of the largest neurite from the cell body to the growth cone. To quantify the number of cells, images of five different fields were taken at random (20× magnitude) in each well. The number of DAPI+ and DAPI+/SATB2+ cells was counted in each photo and the percentage of DAPI+/SATB2+ cells to DAPI+ was extracted.

### 2.7. Histology

Nissl staining (cresyl-violet) was used to quantify the corpus callosum area of animals. After fixation, the brains were split at the midline and sectioned parasagittally in the vibratome (150 μm). The sections were then stained with cresyl-violet, (Sigma-Aldrich, Burlington, MA, USA) for 8 min, then dehydrated with ethanol at different concentrations (75%, 95%, and 100%) for 5 min each. After dehydration, the sections were immersed in 100% xylol (Sigma-Aldrich, Burlington, MA, USA) for 10 min, and the slides were cover-slipped with Entellan.

### 2.8. Dye Labelling

DiI (1,1′-dioctadecyl-3,3,3′,3-tetramethyl-indocarbocyanine perchlorate, Invitrogen) is a lipophilic dye used as an axonal tracer for fixated tissue. After tissue collection and fixation, a DiI crystal was inserted into the pial surface of the dorsolateral parietal cortex of E18 embryos and incubated for 2 weeks in PBS for diffusion from the cell body to the fibers. After incubation, the brains were serially sectioned in a vibratome (150 μm), at the coronal plane, and analyzed under a fluorescence Zeiss Axioimager Microscope.

### 2.9. Isotropic Fractionator for Total and Neuronal Cell Counting

To assess the total number of cells, the isotropic fractionator technique was performed as described by Valério-Gomes et al., 2018 [24]. Briefly, the brains of P4 mouse pups were collected and fixed in 4% PFA by immersion for two weeks. Then, the cerebral cortex was dissected, weighed, and mechanically dissociated for 15 min using a glass tissue homogenizer (Pyrex^TM^, Rosemont, IL, USA) with a buffer detergent solution (40 mL sodium citrate +1% Triton X-100). A suspension of the nuclei was transferred to 15 mL falcon tubes and then prepared by adding 0.1 M of phosphate buffer saline (PBS) until reaching a 5 mL final volume. From this homogeneous suspension, 1 mL 5 aliquots were taken and 2% of DAPI (20 mg/L) was added for total cell quantification in the Neubauer chamber using an Axioimager fluorescence microscope (Zeiss, Tokyo, Japan). The total number of cells was estimated by multiplying the mean nuclear density by the total suspension volume and 15.625 (Neubauer factor number for 16 squares counted). Pyknotic nuclei were counted in parallel with total DAPI nuclei by observations of the presence of small and degenerated DAPI+ nuclei. From this same suspension, other aliquots of 1 mL were taken for immunocytochemistry. To identify and count callosal neurons, immunocytochemistry was performed using the nuclear SATB2 antibody. For that, aliquots were centrifugated and washed trice in 0.1 M PBS (1500 RPM, 5 min each) and then incubated with 1:200 mouse anti-SATB21:200 (GenWay 20-372-60065) in blocking solution (2 μL BSA 5%, 30 μL NGS and PBS to a complete volume of 200) for up to 48 h in agitation at 4–8 °C. After three centrifugations, the aliquots were incubated with 1:500 Alexa 546 anti-mouse in the same blocking solution in agitation at room temperature for 2 h. To estimate the SATB2 absolute number, the percentage obtained by counting SATB2+ positive nuclei in 500 DAPI+ nuclei was multiplied by the total number of cells.

### 2.10. Statistical Analysis

Statistical analysis was performed using GraphPad Prism 8. In vivo, experimental data were analyzed using an unpaired Student’s *t*-test. For in vitro experiments, statistical analysis was considered as paired. Data are expressed as the mean ± S.D. Details for each result were described individually in the results section. For each independent experiment, at least 3 animals were used per group. The sample size was based on previous studies of our group [16,25].

## 3. Results

### 3.1. Congenital ZIKV Infection Leads to a Reduction in the Corpus Callosum in Mice

To investigate the cellular mechanisms underpinning corpus callosum developmental abnormalities after ZIKV infection, embryonic mice were inoculated with 10^4^ PFU of Brazilian ZIKV (ZIKV) or culture medium (MOCK) at E15 in utero via an intracerebroventricular approach and harvested P4, when they were euthanized and had their brains removed (Figure 1A,B). To characterize this infection model, we measured the brain weight at P4 and found that the infected brains were lighter than the controls (Figure 1C). Also, when measuring only the cerebral cortex weight, the effect was evident (Figure 1D). Using ZIKV NS1 immunostaining, ZIKV protein was detected in the cerebral cortex at P4 (Figure 1E,F). Also, a plaque assay confirmed the presence of infectious viral particles in cortical samples (Figure 1G). To analyze the corpus callosum formation, we compared the sagittal brain sections of infected animals to the controls and found that the corpus callosum area is severely reduced in infected animals (Figure 1H–J). Corpus callosum dysgenesis could be caused by multiple developmental defects. To investigate whether the in utero infection impacted the number of callosal neurons, we used the isotropic fractionator method to estimate the total amount of cortical SATB2+ cells (Appendix A). At P4, we observed lower counts of both the total nuclei (DAPI+) and SATB2+ cells when compared to the controls (Appendix A). Together, these results show that 9 dpi animals present abnormal corpus callosum development, with a significant reduction in SATB2+ callosal cells, which is consistent with the callosal dysgenesis found in some patients with congenital Zika syndrome [7].

### 3.2. Congenital ZIKV Infection Causes Cell Death at the Cortical Plate

ZIKV infection is known to cause cell death, contributing to the microcephalic phenotype [2,3,4]. To verify if in our model cell death could account for the SATB2+ callosal cell reduction, we stained the cerebral cortex with cleaved caspase-3 (CASP3+) apoptotic marker at different time points. Interestingly, after 5 dpi (50 PFU, low titre infection), there were no CASP3+ cells in the cerebral cortex (Figure 2A–C). Cell death was only noticed after 7 dpi (Figure 2D–F), and, more pronouncedly, at 10 dpi (Figure 2G–I). In addition, animals infected with ZIKV presented an increased number of pycnotic nuclei at P4 (Appendix A), confirming an increased cell death after 10 dpi. These results suggest that the loss of cortical cells at P4 (Appendix A) could be due to cell death induced by ZIKV infection.

### 3.3. ZIKV Infection Leads to a Defect in Callosal Neuron Production

In addition to cell death, ZIKV is also known to impair the neural progenitor cell cycle [4]. In the low titer ZIKV infection model, cell death was only observed after birth (Figure 2D–F). Therefore, we investigated whether ZIKV embryonic infection would impair callosal neuron production by impacting the proliferation of neural progenitors. At E15, cortical layer V neurons migrate toward the cortical plate and callosal II/III neurons are generated. After 3 dpi (E18), although infected brains show SATB2+ neurons positioned similarly to the controls (Figure 3A–C), their density is significantly reduced (Figure 3D). The cortical plate was divided into three sectors of equal areas encompassing the whole cortical thickness, and SATB2+ neurons were quantified to DAPI+ nuclei. To quantify the progenitor density, we immunostained sections with the PH3 G2/M marker. We observed that ZIKV infection reduces the density of proliferating cells at the ventricular zone after 3 dpi (Figure 3E–G). After 5 dpi, the intermediate progenitor density (TBR2+) is also reduced (Figure 3H–J). Taken together, these results show that ZIKV infection impairs proliferation and reduces intermediate progenitor cells, responsible for generating upper granular layers, where the majority of callosal cells are found. Also, ZIKV infection reduces the callosal neurons’ density within the cortical plate, but not their location (Figure 3D).

### 3.4. ZIKV Infection Impairs Axonal Growth of Callosal Neurons

Axon growth is a critical step in corpus callosum development [13]. To establish whether ZIKV infection directly impairs the axon extension of callosal neurons, we used the E14 cortical neuron primary culture and infected with ZIKV compared to the controls. After 4DIV, no difference was observed in the number of SATB2+ neurons (Figure 4D–F) or the presence of cell death (Figure 4G–I). However, we quantified the length of callosal axons using ImageJ tracing tools and found that they were ~25% shorter compared to the mock exposed axons (Figure 4A–C). To verify whether ZIKV affects callosal axon growth in vivo, a DiI crystal, a lipophilic dye that diffuses within fixated membranes, was inserted and incubated into the dorsolateral cerebral cortex surface to allow for anterograde axonal tracing (Figure 5A–F). After 20 days of DiI diffusion, the E18 brain was sectioned, and axons could be analyzed. Interestingly, infected animals show not only a reduction in the callosal bundle at the midline, but also signs of axonal misrouting (Figure 5J–K). To accurately measure the width of the callosal axons, we stained axons with L1CAM and analyzed the corresponding rostrocaudal levels at 5 dpi. The infected animals display a reduction in the midline bundle width compared to the controls (Figure 5G–I).

### 3.5. ZIKV Infection Impairs Midline Glia Cells

Our in vitro results show that ZIKV infection generates callosal neurons with shorter axons. We also observed that in vivo, axons that are crossing the midline through the corpus callosum are reduced and misrouted. These findings suggest that axon guidance could be altered. The midline glia consists of transient cell populations that hold the role of guiding axons across the midline during development. We measured the signal intensity of GFAP+, a marker of glial cells (Figure 6A–D). In ZIKV-infected mice, GFAP+ intensity is low, suggesting that this cell population is compromised, which may explain the defasciculated axons we observed. Some neurons that compose the midline glial cells at the indusium griseum express calbindin as a marker (Figure 6E,F). In ZIKV-infected animals, this cell population is also reduced (Figure 6G). Also, microglia staining for IBA1+ presents an activated morphology at the cortical region after 7 days of in utero infection (Figure 6H–L). However, quantification at the midline region (Figure 6J–K) was difficult to calculate.

## 4. Discussion

Congenital Zika syndrome is a set of birth defects caused by ZIKV infection during pregnancy, with microcephaly as its hallmark. ZIKV studies mainly focus on the viral impact on neurogenesis and its relation to microcephaly. Nevertheless, other brain abnormalities are present in CZS, such as corpus callosum dysgenesis. In this study, we identified defects in the proliferation and axonal extension of callosal cells, as well as alterations in midline glial cells, which can contribute to our understanding of the cellular mechanisms by which ZIKV infection leads to the malformation of the corpus callosum.

We used in utero infection to assess the brain abnormalities caused by ZIKV. Like other in vivo studies that used this approach, ZIKV was capable of replicating in the mouse cerebral cortex and causing microcephaly (Figure 1A–C) [26,27]. Alongside microcephaly, we found that the corpus callosum size was significantly reduced in infected animals (Figure 1H–J). This result is consistent with reported cases of CSZ in humans [7]. In addition, we found an important reduction in the total number of cells in the cerebral cortex, which agrees with reduced cortical thickness found in other studies of embryonic ZIKV infection [18,26,28,29]. Moreover, SATB2+ cells in particular, which are known to represent cortical callosal projection neurons, were severely reduced in number (Appendix A). The significant change in the number of callosal neurons explains, at least partially, the reduced size of the corpus callosum in infected animals.

ZIKV infection is associated with massive cell death ratios on the cortical plate at 10 dpi (Figure 2C–F). This effect could account for the loss of callosal neurons at P4 (Appendix A). As previously shown, extensive neuronal death is detected in different mouse models of infection [17,26,27,30]. In these models, cell death appears to be concentrated in the cortical plate and takes place both for immature and mature neurons. In our study, we also observed the same phenomenon.

ZIKV infection impairs neurogenesis by disrupting the cell cycle and reducing proliferation [3,4]. In this study, we observed that apoptosis typically occurred in the cortical plate only after birth, therefore we asked if ZIKV infection disrupts callosal neurons during the embryonic period by disturbing neural progenitor cell proliferation. At E15, callosal neurons that occupy cortical layers II/III originate from progenitors located at the ventricular surface of the cerebral cortex [31]. We observed that infecting animals at this age reduces the number of SATB2+ neurons at E18 (Figure 3A,B), a time point at which cell death was not noticed (Figure 2A). In agreement with previous studies [26,27,30], we also found that ZIKV infection reduces proliferation at the ventricular zone, which leads to less intermediate progenitors (TBR2+) (Figure 3D–G). In later stages of corticogenesis, TBR2+ progenitors are responsible for producing most glutamatergic projection neurons in all cortical layers [32]. Therefore, in our model of infection, ZIKV reduces the number of callosal neurons by targeting proliferation and intermediate progenitor cells without causing cell death at the embryonic stage.

Microcephaly is mainly caused by a reduction in the number of neurons due to impaired cell proliferation [33]. Although a reduction in the callosal commissure may be directly caused by this key feature of microcephaly, in mice, we found that corpus callosum malformation is better explained by the disruption of developmental steps after neurogenesis, such as defective midline patterning, axonal misrouting, and growth [14,15]. To evaluate if ZIKV directly impairs the axonal extension of callosal neurons, we infected primary cell cultures of embryonic mice at E14. As observed at 3 dpi, callosal neurons exhibit shorter axons compared to the controls (Figure 4A–C). Our results agree with those of Goodfellow and collaborators (2018) [34], who infected cultured mature neurons with ZIKV and found that neurite length and branching were reduced in size. Here, we show that embryonic SATB2+ callosal neurons have a shorter axon and less arborized dendrites. In addition, using an axonal tracer to analyze callosal tracts in vivo, we found axonal misrouting at the midline (Figure 5K). These results indicate that ZIKV infection compromises an important step of corpus callosum formation.

Axonal misrouting could be explained by the lack of axonal guidance signaling, which is provided by midline glial cells [14,15]. In mouse models of ZIKV infection, glial cells are also disrupted. Li and collaborators (2018) [18] noticed that mice infected at E15 and evaluated at P5 showed a reduced number of oligodendrocytes in the corpus callosum. Here, we observed that during development, infected animals showed a reduction in GFAP+ midline astrocytes (Figure 6A–D), suggesting that these cell populations are impaired, and therefore indicating that the virus is capable of infecting different cell populations in the brain. However, at embryonic stages, only a few regions, such as at the midline, display specialized glia. Later, at postnatal stages, when gliogenesis is at its peak, GFAP is increased in infected animals, indicating reactive gliosis [18,28].

An important question of ZIKV infection is whether the observed effects are caused directly by the viral infection and not derived from inflammatory processes. In our model of ZIKV infection, we observed a massive cell death after 10 dpi (Figure 2C–F) that accounts for the loss of SATB2+ neurons and reduction in the corpus callosum. When we look at the effect of ZIKV infection at neurogenesis at a time point when apoptosis is not observed, this infection causes a reduction in proliferation and callosal neurons but is minimal when compared to the effect caused by cell death. ZIKV infection produces a strong inflammatory response and during embryonic development, inflammatory molecules are known to contribute to neurodevelopmental disorders [35]. Other studies investigating the effect of ZIKV infection on gene expression demonstrated an upregulation of genes that are involved in viral infection, and immune and inflammatory response [26,28,30]. Also, we found elevated activated microglia in infected animals (Figure 6E–I). Infection of microglia can result in the dissemination of ZIKV through the brain, as well as the production of pro-inflammatory cytokines [19,36]. These results show that neuroinflammation contributes to brain abnormalities caused by ZIKV infection.

Despite the strong presence of inflammatory factors, ZIKV infection also deregulates the key genes important for neurodevelopment [25,37]. Semaphorins and Ephrins are a family of proteins that act on axonal guidance by attracting or repelling growing axons at the midline [38,39,40]. The reduction in the GFAP+ marker of midline glia could account for lower expression levels of cue molecules and may generate a poor signaling environment for crossing axons at the midline. Also, processes involving axon and dendrite development were found to be deregulated in previous work [37].

As the formation of the corpus callosum comprises complex steps of development, this is the first study to suggest a mechanism by which ZIKV infection interferes with the formation of the corpus callosum during embryonic development. These findings contribute to our understanding of the pathophysiology of ZIKV infection. CSZ patients exhibit a range of brain abnormalities that are not always associated with microcephaly [41]. The ability of ZIKV to disturb critical steps of corpus callosum development reveals the importance of monitoring pregnant women who have been exposed to ZIKV and doing a follow-up of their newborns since the corpus callosum continues to develop postnatally [42].

Further studies should address whether the changes in axonal extension and axon guidance can be explained by the lack of key players that are crucial for cortical development, either produced by the midline glia, which is altered in our work, or by other cells that also play this role.

## Figures and Tables

**Figure 1 viruses-15-02336-f001:**
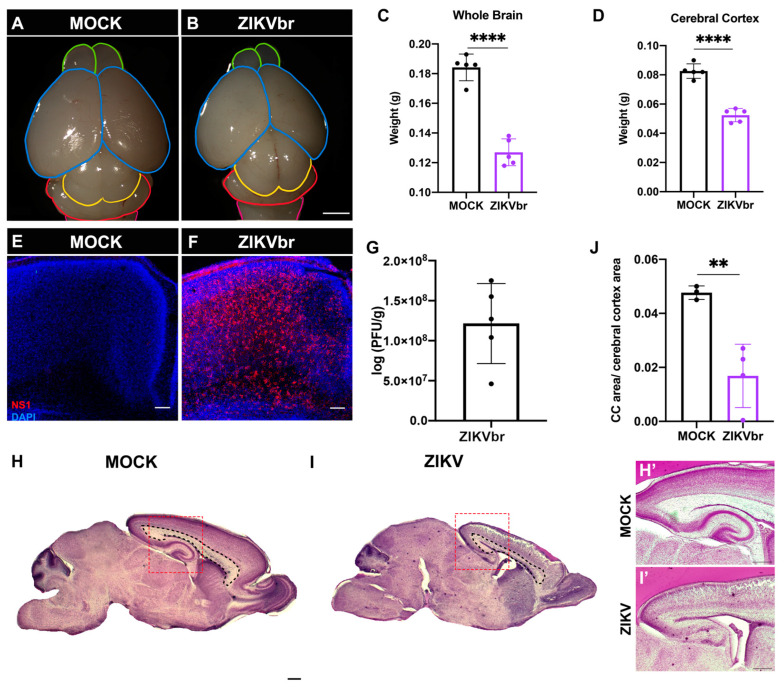
Zika virus congenital infection at E15 causes microcephaly and corpus callosum dysgenesis at P4. (**A**,**B**) Dorsal views of MOCK animals and ZIKV-infected mice. Scale bar = 1 mm. Colored lines delimit the anatomical structures of the mouse brain. Olfactory bulb (green); cerebral hemispheres (blue); midbrain (yellow); cerebellum (red); medulla (pink). (**C**,**D**) Quantitative analysis of the whole brain (unpaired *t*-test, t = 10.06, df = 8, *p* < 0.0001) and cerebral cortex (unpaired *t*-test, t = 10.00, df = 8, *p* < 0.0001) weights (g) of the ZIKV and MOCK groups. N= MOCK (5) ZIKV (5). (**E**,**F**) Immunohistochemical labeling for NS1 (red) and DAPI staining (blue) on MOCK and ZIKV-infected cerebral cortex. Scale bars = 50 μm (**G**) Quantitative analysis of infected particles in ZIKV animals by a plaque assay. N = ZIKV (5). (**H**,**I**) Sagittal sections of MOCK and ZIKV-infected brains, stained with cresyl-violet. The corpus callosum area is delimited by black dashed lines. Red dashed lines represent the area of higher magnification of the corpus callosum of MOCK (**H’**) and ZIKV-infected (**I’**) brains. Scale bar = 100 μm. (**J**) Quantification of the corpus callosum area normalized to the cerebral cortex area in quasi-sagittal brain sections of ZIKV and MOCK animals. (Unpaired *t*-test, t = 4.381, df = 5, *p* = 0.0071) N = MOCK (3) ZIKV (4). ** *p* ≤ 0.01, **** *p* ≤ 0.0001.

**Figure 2 viruses-15-02336-f002:**
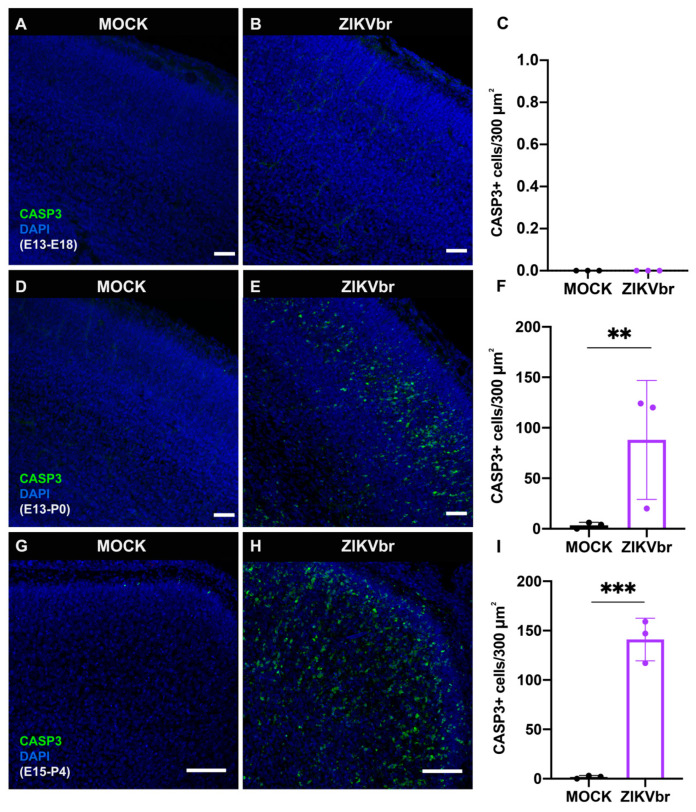
Zika virus congenital injection results in cell death only after 7 days post infection. (**A**,**B**) Immunohistochemistry for activated Caspase 3 (CASP3) on coronal sections of MOCK compared to ZIKV-infected E13 embryos, harvested at E18. DAPI staining in blue. (**C**) Quantification of CASP3+ cell density in 300 μm^2^ at E18. N = MOCK (3) ZIKV (3). (**D**,**E**) Immunohistochemistry for CASP3 (green) on coronal sections of MOCK compared to ZIKV-infected E13 embryos, harvested at P0. Scale bars = 50 μm. (**F**) Quantification of CASP3+ cell density in 300 μm^2^ at P0. (Unpaired *t*-test, t = 6.187, df = 4, *p* = 0.0035) N = MOCK (3) ZIKV (3). (**G**,**H**) Immunocytochemistry for CASP3 (green) on coronal sections of MOCK compared to ZIKV-infected E15 embryos, harvested at P4. Scale bars = 50 μm. (**I**) Quantification of CASP3+ cell density in 300 μm^2^ at P4. (Unpaired *t*-test, t = 11.13, df = 4, *p* = 0.0004) N = MOCK (3) ZIKV (3). ** *p* ≤ 0.01, *** *p* ≤ 0.001.

**Figure 3 viruses-15-02336-f003:**
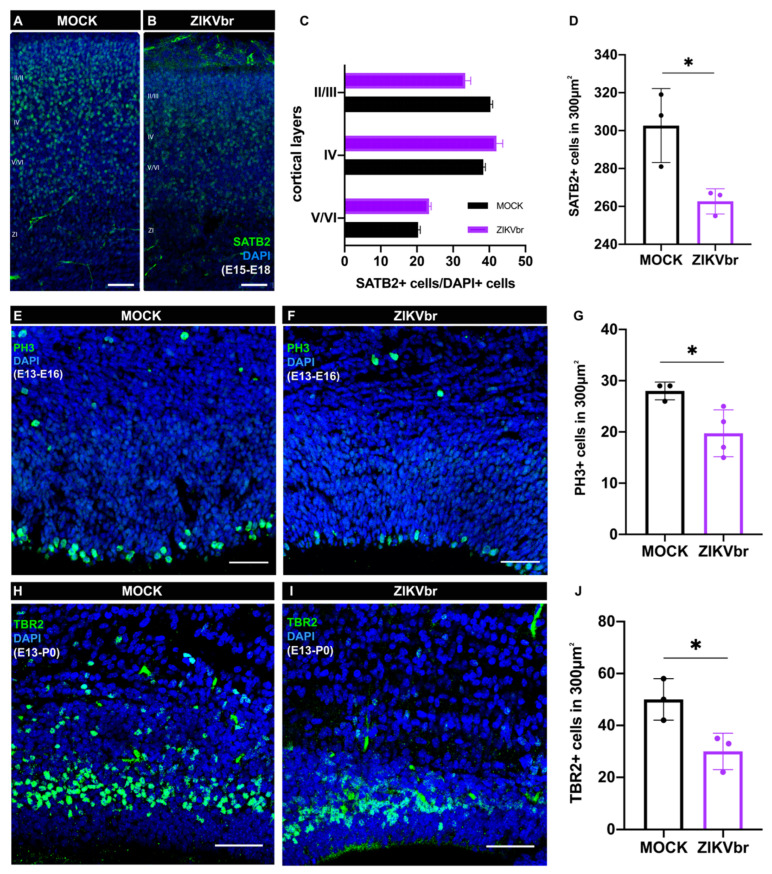
Zika virus in utero injection leads to intermediate progenitors and callosal neuron density reduction. (**A**,**B**) Immunohistochemistry for callosal nuclear marker SATB2 (green) counterstained with DAPI (blue) in cortical coronal sections of MOCK and ZIKV mice injected in utero at E15 and harvested at E18. (**C**) Quantifications of SATB2+/DAPI+ cells in the different cortical layers (I to VI). N= MOCK (3) ZIKV (3). (**D**) Quantifications of SATB2+ cell density in 300 μm^2^. (Unpaired *t*-test, t = 4.381, df = 5, *p* = 0.0285) N = MOCK (3); ZIKV (3). (**E**,**F**) Immunohistochemistry for phospho-histone 3 (PH3) of MOCK compared to ZIKV-infected E13 embryos, harvested at E16 in 300 μm^2^. (**G**) Quantification of PH3+ cell density in 300 μm^2^. (Unpaired *t*-test, t = 2.913, df = 5, *p* = 0.0333) N = MOCK (3); ZIKV (3). (**H**,**I**) Immunohistochemistry for intermediate progenitors’ marker TBR2 (green) in the cortical sections of MOCK and ZIKV-infected E13 embryos, harvested in P0. (**J**) Quantification of TBR2+ cell density in 300 μm^2^. (Unpaired *t*-test, t = 3.259, df = 4, *p* = 0.0311) N = MOCK (3); ZIKV (3). * *p* ≤ 0.05 Scale bars = 50 μm.

**Figure 4 viruses-15-02336-f004:**
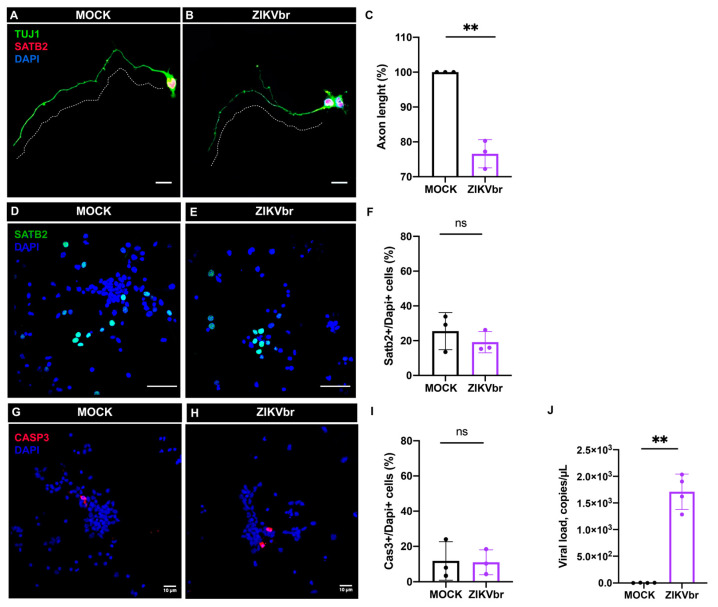
Zika virus in vitro infection reduces axon extension and dendritic arborization of callosal neurons. (**A**,**B**) Primary dorsal cortical neurons were exposed to ZIKV or MOCK after 24 h and cultivated for 3 days post infection (3 dpi). Co-immunolabelling for TUJ1 (green) and SATB2 (red) to measure the longest neurite (dashed lines). Scale bars = 10 μm. (**C**) Quantification of axonal length in ZIKV and MOCK primary neuron cultures. (Paired *t*-test, t = 10.01, df = 2, *p* = 0.0098) N = 3 independent experiments. (**D**,**E**) Primary cortical neurons exposed to ZIKV or MOCK after 24 h and cultivated for 3 dpi. Immunolabelling for SATB2 (green). Scale bars = 50 μm. (**F**) Quantification of SATB2 neurons in relation to DAPI in ZIKV and MOCK primary neuron cultures. (Paired *t*-test, t = 1.073, df = 2, *p* = 0.3954) N= 3 independent experiments. (**G**,**H**) Primary dorsal cortical neurons exposed to ZIKV or MOCK after 24 h and cultivated for 3 dpi. Immunolabelling for Caspase-3 (red). Scale bars = 10 μm. (**I**) Quantification of +CASP-3/DAPI in ZIKV and MOCK primary neuron cultures. (Paired *t*-test, t = 0.3236, df = 2, *p* = 0.7770) N = 3 independent experiments. (**J**) ZIKV viral RNA was determined using real-time PCR after 3 dpi from four independent experiments. ns = not significant, ** *p* ≤ 0.01(Paired *t*-test, t = 10.30, df = 3, *p* = 0.0020).

**Figure 5 viruses-15-02336-f005:**
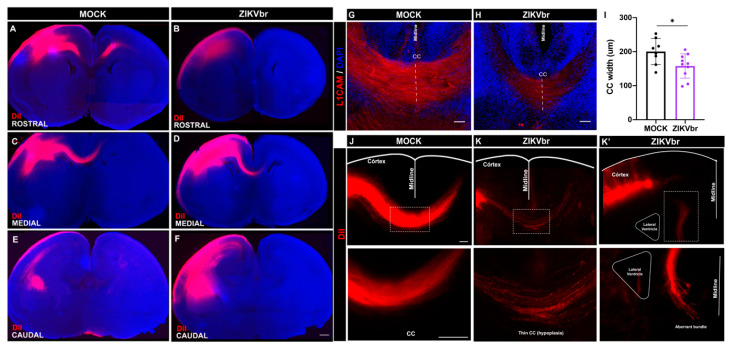
Zika virus congenital infection impairs the midline crossing of callosal axons at E18 in vivo. (**A**–**F**) Rostro-caudal levels of DiI anterograde tracer (red) in E18 brains of MOCK and ZIKV infected animals at E13. Scale bar = 100 μm. (**G**,**H**) L1CAM labeling of callosal axons at midline region of MOCK or ZIKV infected animals at E13 and harvested at E18. Scale bars = 50 μm. (**I**) Width measurement of callosal axons labeling for L1CAM (red) at the midline region. Blue = DAPI. (Unaired *t*-test, t = 2.434, df = 16, *p* = 0.0270) N = MOCK (8); ZIKV (10). (**J**,**K**) Brain coronal section of MOCK animal with DiI labeling fibers compared to the ZIKV-infected animal with defasciculated fibers at midline. Scale bars = 50 μm. (**K’**) Brain coronal section of ZIKV-infected animal with DiI labeling fibers showing misrouted axon fibers that fail to cross the midline. The dashed line corresponds to a magnification of the corpus callosum area. Scale bar = 50 μm. CC = corpus callosum Zika virus in vitro infection reduces axon extension and dendritic arborization of callosal neurons. * *p* ≤ 0.05.

**Figure 6 viruses-15-02336-f006:**
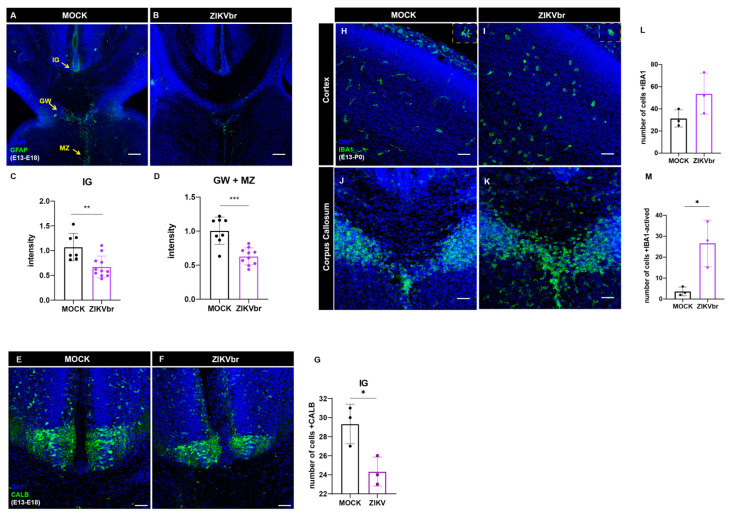
Zika virus infection at E13 alters midline glial populations and leads to microglial activation. (**A**,**B**) Immunocytochemistry for GFAP (green) at the midline regions of embryonic brain slices of MOCK (**A**) and ZIKV (**B**) animals infected in utero at E13 and analyzed at E18. (**C**) Quantification of GFAP signal intensity at the indusium griseum glia (IG) close to the midline of ZIKV and MOCK. (Unaired *t*-test, t = 3.941, df = 16, *p* = 0.0012) N = MOCK (8); ZIKV (10). (**D**) Quantification of analysis of GFAP signal intensity at the midline zipper glia (MZ) + glial wedge (GW) region of ZIKV and MOCK. (Unaired *t*-test, t = 4.897, df = 16, *p* = 0.0002) N = MOCK (8) ZIKV (10). (**E**,**F**) Immunocytochemistry for calbindin (CALB) (green) at the indusium griseum glia (IG) close to the midline of MOCK and ZIKV infected in utero at E13 and analyzed at P0. (**G**) Quantification of CALB+ density at the indusium griseum glia (IG) close to the midline of ZIKV and MOCK. (Unaired *t*-test, t = 3.354, df = 4, *p* = 0.0285) N = MOCK (3); ZIKV (3). (**H**–**K**) Immunocytochemistry for microglial marker IBA1 (green) at the cortical plate of embryonic brain slices of MOCK (**H**) and ZIKV, (**I**) and the midline regions of MOCK (**J**) and ZIKV (**K**) infected in utero at E13 and analyzed at P0. (**L**) Quantification of IBA1+ density at the cortical plate in 450 μm^2^. (Unaired *t*-test, t = 1.923, df = 4, *p* = 0.1268) N = MOCK (3); ZIKV (3). (**M**) Quantification of IBA1+ density at the cortical plate with amoeboid morphology (active microglia) in 450 μm^2^. (Unaired *t*-test, t = 3.540, df = 4, *p* = 0.0240) N = MOCK (3) ZIKV (3). Scale bars = 50μm. Blue = DAPI. IG = indusium griseum glia; MZ = midline zipper glia; GW = glial wedge. * *p* ≤ 0.05, ** *p* ≤ 0.01, *** *p* ≤ 0.001.

## Data Availability

The data presented in this study are available on request from the corresponding author.

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
