# Peer review of "Congenital Zika Virus Infection Impairs Corpus Callosum Development"

_viruses, 2023, doi:10.3390/v15122336_

Round 1

Reviewer 1 Report

Comments and Suggestions for Authors

Comments on the Quality of English Language

There are some spelling/grammar/formatting errors in the current version of the manuscript, please correct them carefully.

Reviewer 2 Report

Comments and Suggestions for Authors

In the article by Raissa et al., the authors investigated  how ZIKV infection leads to callosal malformation. The questioned raised is of interest however, the quality of their findings specifically the images in Figures 1 A and B, Figure 4, Figure 5 and Figure 6 E and F, are not convincing. In Figures 1 A and B, the cortex appears smaller in ZIKV but the length of the cerebellum appears longer when compared to MOCK.  Moreover, it appears that Figure 5 C and E have been altered.  For that I vote to reject this article. 

Other minor concerns include clarity in the methodology regarding the animal infection paradigm, maybe provide a schematic illustration. Line 87: 1.5 uL is equal to 104PFU or 10PFU.? More detail how DIL was administered  to E18 embryos in utero.  How many E and P brains were harvested ?

Comments on the Quality of English Language

Minor editing of English language required

Reviewer 3 Report

Comments and Suggestions for Authors

In this manuscript, the authors present compelling evidence of developmental defects in the corpus callosum (CC) of a mouse model resulting from ZIKV infection. This finding is particularly significant due to its relevance to schizophrenia (SCZ) patients. The authors have made a concerted effort to shed light on the underlying mechanisms through the examination of several key aspects:

1.      Widespread neural cell death.

2.      Impaired neural progenitor proliferation, reduced numbers of intermediate progenitors, and satb2+ neurons.

3.      Reduction and misrouting of callosal axons.

4.      Altered characteristics of glial cells, suggesting changes in axonal guidance signaling.

While the study is commendable, the paper has raised two critical questions that merit further investigation:

1.      Notably, the paper reports a decrease in the number of GFAP-positive cells, which contrasts with other research suggesting an increase in GFAP numbers in response to similar conditions, for example Huang W C, Abraham R, Shim B S, et al. Zika virus infection during the period of maximal brain growth causes microcephaly and corticospinal neuron apoptosis in wild type mice[J]. Scientific reports, 2016, 6(1): 34793. We suggest that the authors explore potential factors contributing to this discrepancy, such as variations in the timing or brain region of measurements. Additional research is needed to resolve this discrepancy.

2.      The paper highlights an increase in the number of microglia cells within the corpus callosum, which contrasts with the resulting decreased thickness of this structure. The authors should consider factors that might account for this paradox. It is important to carefully explain this phenomenon.

In conclusion, this manuscript has raised intriguing questions about the effects of ZIKV on neural development, but further research is essential to unravel the underlying mechanisms and clarify any experimental variations. The authors are encouraged to address these questions and consider them in the context of their study.

Round 2

Reviewer 1 Report

Comments and Suggestions for Authors

All the concerns have been addressed by the authors.

Reviewer 2 Report

Comments and Suggestions for Authors

I am happy with the changes made by the authors.  The differences between mock and viral-infected brains in Fig. 5 and Fig.6 are noticeable. 

Comments on the Quality of English Language

I am happy with the changes made by the authors.  The differences between mock and viral-infected brains in Fig. 5 and Fig.6 are noticeable. I accept the revised manuscript in its present form. 

Reviewer 3 Report

Comments and Suggestions for Authors

Accept in present form